# Hepatoid Adenocarcinoma of the Lung: A Review of the Most Updated Literature and a Presentation of Three Cases

**DOI:** 10.3390/jcm12041411

**Published:** 2023-02-10

**Authors:** Alessandro Bonis, Andrea Dell’Amore, Vincenzo Verzeletti, Luca Melan, Giovanni Zambello, Chiara Nardocci, Giovanni Maria Comacchio, Federica Pezzuto, Fiorella Calabrese, Federico Rea

**Affiliations:** 1Thoracic Surgery Unit, Department of Cardiac, Thoracic, Vascular Sciences and Public Health–DSCTV, University of Padova, 35122 Padova, Italy; 2Pathology Unit, Department of Cardiac, Thoracic, Vascular Sciences and Public Health–DSCTV, University of Padova, 35122 Padova, Italy

**Keywords:** hepatoid adenocarcinoma, hepatoid adenocarcinoma of the lung, HAL, rare lung cancer

## Abstract

In a Surgical Thoracic Center, two females and a man were unexpectedly diagnosed with hepatoid adenocarcinoma of the lung (HAL) in a single year. HAL is a rare lung cancer with pathological features of hepatocellular carcinoma with no evidence of liver tumor or other primitive sites of neoplasms. As of today, a comprehensive treatment is still not written. We reviewed the most updated literature on HAL, aiming to highlight the proposed treatments available, and comparing them in terms of survival. General hallmarks of HAL are confirmed: it typically affects middle-aged, heavy-smoker males with a median of 5 cm bulky right upper lobe mass. Overall survival remains poor (13 months), with a longer but non-significant survival in females. Treatments are still unsatisfactory today: surgery guarantees a small benefit compared to non-operated HALs, and only N0 patients demonstrated improved survival (*p* = 0.04) compared to N1, N2, and N3. Even though the histology is fearsome, these are probably the patients who will benefit from upfront surgery. Chemotherapy seemed to behave as surgery, and there is no statistical difference between chemotherapy only, surgery, or adjuvant treatments, even though adjuvant treatments tend to be more successful. New chemotherapies have been reported with notable results in recent years, such as Tyrosine Kinase Inhibitors and monoclonal antibodies. In this complicated picture, new cases are needed to further build shared evidence in terms of diagnosis, treatments, and survival opportunities.

## 1. Introduction

Lung cancer is the second most frequent neoplasm in both sexes and the first cause of tumor-related death [1]. Because of its frequency, lung cancers presenting rare histology remain understudied [2]. Among them, a particular interest is reported in hepatoid adenocarcinoma of the lung (HAL). Hepatoid adenocarcinoma (HAC) demonstrates morphological and microscopical features of hepatocellular carcinoma (HCC), having no evidence of primitive liver malignancy [3]. HAL involvement accounts for only 2.5% of all cases of HAC, and it is less frequent than other sites, such as the stomach and ovary (respectively, 63% and 10%) [4,5]. Considering this, it is also necessary to exclude the possibility that other organs, more commonly affected by HAL, have metastasized to the lung.

In addition to the pathological details, the tumor microenvironment has also been little studied. It is a complicated net of signal pathways and molecular interactions that promote cancer development and protect the neoplasm. In HAL, this is still today underexplored: EGFR, ALK, RAS, ROS, MET, and PDL-1 are routinely processed to discover any therapeutical opportunities in lung adenocarcinomas (ADCs), and this also happens in HALs [6]. Recently, a new driver gene was introduced—a cadherin-coding gene called FAT1 [7]. When expressed, FAT1 demonstrated a more frequent response to immunotherapy, and this gene is also involved in the NSCLC microenvironment (interaction with the Hippo pathway) [8,9]. FAT1 was also evaluated in HCCs and in ADCs: it was expressed in both of them, and it was mutated in almost 12% of ADCs (especially a missense mutation) and in 5% of HCCs (more frequently deleted). Moreover, this mutation demonstrated a drug sensitivity to TKIs (Gefitinib and Afatinib) [10].

The first description of HAL is dated 1981 and appeared in a scientific report of a patient presenting with lung cancer, high levels of alpha-fetoprotein (aFP), and no evidence of liver disease [11]. In 1990, it was finally defined as HAL [12]. An elevated aFP is reported in the literature as a surveillance and diagnostic biomarker in HCC, but its diagnostic role in HAL remains unsure [13,14,15,16].

Since 1990, the published case series and case reports have fixed almost three landmarks in HAL. First, a strong male predominance. To date, HAL is found to have a low overall incidence, but in females, it is extremely rare. Male predominance amounts to approximately 80% of cases [3,17,18]. Haninger et al. described the first case of a woman’s HAL who incredibly survived almost 9 years after diagnosis [19]. Curiously, females appear to live longer than men do, probably protected by unknown mechanisms. Hormones or gender roles in HAL remain a still open discussion [10].

Secondly, in pathological reports, HAL is commonly diagnosed with panels showing positivity for HepPar1, aFP, CK7, and pancytokeratin. TTF1, napsin, and CK20 are commonly negative. Immunohistochemical stains and microscopic features of HAL have been previously largely discussed [3,10,20,21].

Finally, patients usually receive a diagnosis of advanced oncological disease, and their expected overall survival is poor. According to previous studies, HAL demonstrates a median overall survival of almost 14–16 months, a 5-year survival rate of 8.0%, and a 2-year survival of 35.3% [10,15,22].

Therefore, rare cancers suffer from a scarcity of background literature and an absence of a general shared consensus treatment. This unfortunately makes the therapy of these patients unhelpful, thus resulting in an oncological behavior still unclear today. In order to increase knowledge on HAL, we summarized and discussed treatments described in the literature as of today, and we reported three cases of HAL (two females and a man). Increasing the number of rare tumors reported encourages further understanding of the pathological mechanisms behind this cancer in the hope of improving better outcomes and treatment agreements in the future.

## 2. Materials and Methods

We interrogated Scopus, EMBASE, MEDLINE, and Cochrane Library online databases, and it was updated to the end of 31 December 2022. To include all the publications dealing with our topic, the query string was composed as (Hepatoid Adenocarcinoma) AND (Lung). We conducted our research according to the updated guidelines for systematic reviews (PRISMA 2020 statement) [23]. Duplicates were manually checked and independently removed by two different authors (A.B. and V.V.). Subsequently, they separately assessed the eligibility of the studies by screening the article’s titles and abstracts and then decided on their inclusion by reading the full text of the selected works. Any discrepancy between authors was solved by consulting the principal investigator (A.D.A.). The inclusion criteria were as follows: (1) English language, (2) publication 31 December 2022, (3) articles reporting at least 1 case of HAL, (4) articles that clearly reported the treatment (surgery or chemotherapy) proposed in HAL. The principal exclusion criteria were: (1) languages other than English, (2) articles that did not contain clear histologic evidence of HAL, (3) lungs as sites of metastatic tumor, (4) conference abstract, editorial material, and letters, (5) studies that omitted radiological, clinical, or pathological data.

We conducted descriptive and survival statistics in Jamovi software (v2.3.21) [24] and R statistical software (v4.2.2–R Core Team 2022) [25]. Categorical variables are explained as numbers and frequencies n(%), and continuous variables are explained as mean ± standard deviation, interquartile range (IQR), and median. We built the reported curves using the ggplot2 R package. We used the Log-rank test to compare them. Statistical significance has been fixed to *p* < 0.05.

Anatomopathological reports collected the following issues. The macroscopical examination described: size, site, and height of the neoplasm and the number and size of the resected lymph nodes. The microscopical examination reported: lymph nodes involvement, microscopical aspect of cells organization, and immunophenotypical results that we reported in a dedicated table (Table 1). Our thoracic specialized pathologists (F.P and F.C) reported in all three cases: (A) hematoxylin and eosin staining, original magnification ×50; (B) hematoxylin and eosin staining, original magnification ×100; (C) IHC: Cytocheratin MNF116, original magnification ×100; (D) IHC: HepPar-1, original magnification ×100.

## 3. Case Presentation

### 3.1. Case 1

A 67-year-old non-smoker female presented to our Thoracic Surgery Department in October 2021, with an advanced oncological status, for which she was being treated in another hospital.

In September 2020, she was admitted to the Emergency Department for left chest pain. A CT scan revealed a 68 mm mass in the upper lobe of the left lung and multiple suspected metastases in both the iliac bones and in the subcutaneous fat of the right buttock. Subsequently, she underwent positron emission tomography (PET) studies which confirmed an increased metabolic activity in the above-mentioned lesions with a maximum standard uptake value (SUV) of 23.2 in the left lung. No other uptake sites emerged, especially in the stomach or ovary. A percutaneous computed tomography-guided biopsy was performed, sampling a lesion in the right ilium. The microscopical examination suggested a poorly differentiated adenocarcinoma with unclear immunohistochemical stain (TTF-1-, CgA-, Syn-, PD-L1 < 1%). She underwent chemotherapy with platinum and gemcitabine for 6 cycles. Unfortunately, a CT scan in May 2021 revealed signs of progression. Therefore, the patient was advised to undergo a second-line therapy with Atezolizumab, which had to be interrupted after one cycle due to toxicity presented by asthenia, amaurosis, drowsiness, and arterial hypotension.

The patient came to our department in poor general health and with dyspnea. An additional CT scan was performed, showing the further progression of the disease, left pleural effusion with pulmonary atelectasis, multiple bone metastases, and the bilateral involvement of mediastinal and hilar lymph nodes (cT4N3M1). We proceeded to investigate by biopsying a 46 mm diameter left axillary lymphadenopathy, which demonstrated a poorly differentiated epitheliomorphic neoplasm with a solid and micro-acinar pattern. This is compatible with HAL (HepPar1+).

Respiratory symptoms were palliated by the placement of a 20Ch pleural drainage and by performing talc pleurodesis (Figure 1 and Figure 2). The patient was then transferred to a palliative care center closer to home. One month after being discharged, brain metastasis occurred, and the patient died of a pulmonary embolism a few days later.

### 3.2. Case 2

In January 2022, a 61-year-old female presented with chest pain and shoulder discomfort. A chest X-Ray and a chest CT scan reported a 65 × 57 mm^2^ mass in the right upper lobe of the lung. The neoplasm highlighted by the CT scan demonstrated a chest wall infiltration between the third and the fifth ribs. It was investigated with a needle biopsy resulting in an initial diagnosis of non-small cell lung cancer not otherwise specified (NSCLC NOS) type. Furthermore, the patient underwent a PET study illustrating an SUV of 16.67 and 6.97 in the mass and in the right paratracheal lymph node station, respectively (cT3N1M0), with no other sites of active glucose metabolism.

The patient underwent 8 cycles of Carboplatinum and Paclitaxel, and a radiation therapy (RT) dose of 50.4 Gy in 28 fractions. At the end of the therapies, restaging images reported the growth of the lung mass (70 × 80 mm^2^) with increased necrotic findings. Given these outcomes, we performed a right upper lobectomy with the removal of the second to the fifth ribs using thoracotomy access and a further wall reconstruction with a Prolene patch (Figure 3 and Figure 4). The final pathological report diagnosed an HepPar1+ lung mass compatible with HAL.

The pathological report diagnosed a poorly differentiated carcinoma with hepatoid morphology (HAL) with a HepPar1+ and a CK7- stain. Moreover, NGS revealed a mutation of exon 11 in the KIT gene (p. N564Y). We returned the patient to our Oncologist for a clinical and radiological follow-up. The patient is still alive 5 months after surgery.

### 3.3. Case 3

A 78-year-old male smoker came to our attention in September 2022 due to a suspected diagnosis of left lower lobe carcinoid. He presented in 2014 a 15 × 10 mm^2^ nodule to the left lower lung that remained substantially unmodified until 2022, when he performed a chest X-ray and a CT scan due to a thoracic trauma, which demonstrated a 33 mm nodule to the left lower lobe. He underwent a bronchoscopy that evidenced a red lesion to the first part of the apical segmental branch. Pulmonologists performed multiple biopsies (TTF1-, p63-, synaptophysin-, pancitocheratines AE1/AE3+) with morphologic and immunohistochemical suspects of adenocarcinoma. A further total body CT scan confirmed the lung as a single site of neoplasm. Moreover, the PET scan did not evidence any significative uptake value. According to this unclear diagnosis, the patient underwent left lower lobectomy and loco-regional lymphadenectomy in Video-Assisted Thoracoscopic Surgery (VATS) (Figure 5 and Figure 6). The pathological report diagnosed a primary lung hepatoid adenocarcinoma (HepPar1+, BRG1+, MNF116+, TTF1-, p40-, chromogranin-, Ki67 60%) with a PDL1 expression of 90%. Pan-TRK panel, EGFR, ALK, ROS, MET, and RET were stained negative. The neoplasia was staged pT1bN3M0 (positive lymph nodes in carina, bronchus intermedius, and left inferior pulmonary vein). We discharged the patient after three days, with no postoperative complications. After fifteen days, we removed the surgical stitches. According to the pathological report, we send the patient to our oncologists to evaluate any possible enrollment to immunotherapy protocol. They proposed an adjuvant treatment with Atezolizumab. After 5 months, the patient is alive with no evidence of disease.

## 4. Results

According to the PRISMA 2020 flow diagram, we identified 50 studies available, with 63 cases of HAL (Figure 7 and Appendix A). In this sample, there were only seven cases of females. We included our 3 patients, with 66 cases of HAL (9 cases of females—13.8%) (Table 2). Demographical, pathological issues, and proposed treatment data are collected in Table 2. HAL cohort evidenced a high tobacco smoking history (44 cases—66.7%), in almost one-third of cases, thoracic pain was encountered at the diagnosis (25 cases—36.8%), there was a common right-side involvement (38 cases—64.4%) and a strong male predominance (57 cases—86.4%). Alpha-fetoprotein (aFP) was available in 41 cases, and when increased, levels were extremely high (median of 902 ng/dL). Considering gender-specific data, women achieved a median age of 65 years (range 53–69) and a mean neoplasm diameter of 5.9 cm (range 1.8–9). In the median, men were younger (60 years old, range 33–79) with a comparable median neoplasm size (5.7 cm; range 2.2–11 cm). Upper fields were commonly involved in both sexes (34 cases–56.7% in men and 7 cases–11.7% in females). Women demonstrated a more frequent left-side involvement (5 cases–8.5%) compared to men, which were commonly affected by right-side cancer (35 cases–59.3%). Three females were in stage III, and in the other four cases, the disease was already metastatic (stage IV); 19 men were in stage III (33.3%), and the other 20 were in stage IV (35.1%). In summary, 82.4% of HALs were in stage III or IV at the diagnosis. Lymph node stations were involved in the diagnosis in 40 cases (72.7%). HALs were metastatic in 23 cases (42.6%).

Considering our experience and the other 48 cases in which authors reported the proposed treatments, HAL was managed as follows: surgery (25 cases—50.0%), chemotherapy (32 cases—64.0% including 7 cases of adjuvant chemotherapy–13.8%) (Table 2). Further, 22 patients (32.4%) were treated in a double (16 cases–24.2%) or triple (6 cases–7.6%) platinum-based chemotherapy, as described in Table 3.

Platinum was adopted in 21 cases (Cisplatinum 10 cases, Carboplatinum 9 cases, and 2 cases of Oxaliplatinum).

Paclitaxel (10 cases) and Docetaxel (3 cases) were the most common drugs used in combination with platinum. Pemetrexed was employed five times. Other drugs combined with the platinum-based regimen were Etoposide, Gemcitabine, Uracil, or Vinorelbine.

Other drugs combined into a first platinum-based scheme were Pembrolizumab (4 cases), Durvalumab, and Bevacizumab (2 cases). Tyrosine Kinase Inhibitors (TKIs) available as first regimen treatments were Anlotinib, Crizotinib, Erlotinib, and Sorafenib.

Some authors also reported a second line or a savage therapy (15 cases—28.3% of treated patients). In six cases, another platinum-based scheme was administrated. Monoclonal antibodies were used in five cases (Atezolizumab, Bevacizumab, and Nivolumab). As concerns TKIs, Anlotinib, Sorafenib, Erlotinib, and Sorafenib were administrated.

Overall survival remained poor, with a median survival of 14 months in the entire cohort. Considering gender, HAL returned an overall survival of 13 months in males and 53 months in females. Women seemed to enjoy a statistically non-significant better survival probability (1- and 3-year survival of 87.5% and 58.3% in females and 52.3% and 26.1% in males, respectively) (Figure 8A). Surgery did not guarantee significance in terms of survival, but operated patients demonstrated a 1-year and 3-year survival of 66.7% and 33.3% versus 60.0% and 27.7% in non-operated ones, respectively) (Figure 8B). Early stages with free lymph stations (N0) evidenced a better survival (1-year and 3-year survival of 60% in N0 compared to a 1-year and 3-year survival of 51.3% and 11.1% in N1, N2, or N3 ones; *p* = 0.04) especially one year after surgery (Figure 8C). Adjuvant treatments gave an advantage in terms of survival, but significance was not statistically highlighted (Figure 8D).

## 5. Discussion

In 1990, HAL was defined for the first time as an independent and extremely rare primitive tumor of the lung [12]. As of today, almost seventy cases of HAL have been described in the literature, empowering a better comprehension of this aggressive neoplasm in the last few years.

Ishikura and Colleagues reported the differential diagnoses that need to be kept in mind when approaching the diagnosis of HAL [12]. Clinicians must always exclude large cell NSCLC, ADCs, and metastatic tumors, especially from the liver, stomach, and ovary. The stomach and ovary must be carefully investigated because they are the most frequent site of hepatoid cancer. HCC must be investigated because the lung can be the site of metastasis. Adenocarcinoma and large cell tumors should be excluded by incidence (much more frequent).

The main hallmarks in HAL remain confirmed. It commonly affects middle-aged men (more than 80%) with a heavy tobacco smoke history. The right upper lung is the main affected field, and it is usually diagnosed in an advanced stage as a bulky mass, and pain is a well-reported alarming symptom [3,15].

Despite the general male predominance of this malignancy, we surprisingly registered two female cases in a single tertiary Center in almost one year, being, to our knowledge, the eighth and ninth cases of female HAL reports. According to Grossman et al. and Hou et al., the patient of Case 2 appeared to be adherent to previous usual HAL features as presenting a right upper lobe mass in a middle-aged smoker with a non-responsive disease after doublet therapy (platinum and paclitaxel) [3,15]. In this case, a feasible demolitive surgical approach and a wall reconstruction permitted the removal of the neoplasia with a complete surgical radicality, according to the anatomopathological report. Five months postoperative, the patient is still alive with no evidence of disease. In Case 1, we describe an unusual HAL presentation: first, she was a non-smoker female. Secondly, she presented with an advanced disease having no surgical options available to the left lower lobe, being a non-responsive disease to common doublet chemotherapy (platinum and gemcitabine), and manifesting an extremely poor prognosis. Thus, a bulky mass, especially affecting the upper lobes in a patient with no response to the usual chemotherapy protocol, must be alarming. As previously discussed, an elevated aFP increases preoperative HAL diagnosis, and it should be sought when HAL is suspected.

As we confirmed, survival in HAL remains scarce. Despite this, it appears to be longer in females compared to males. On this point, a clear statistical significance has not been demonstrated yet. Recently, it has been reported that females exhibit general protection against HAL in comparison to males. However, a clear explanation is unknown [10]. In the future, a better understanding of HAL-related molecular pathways and other female HAL cases could be welcomed.

As concerns treatments, surgery is adopted in about half of HAL cases, but a unique consensus treatment is yet to be outlined. Probably, early stages and N0 HALs demonstrated an interesting, improved survival after surgical treatment because this aggressive histology does not affect cancer development immediately. On this point, the importance of radical surgery in improving outcomes in HAL patients’ survival has already been discussed [15,52]. In Case 3, the patient underwent lobectomy and radical sampling with a poor pathological report (pT1N3M0), but a 90% PDL-1 permitted enrollment into an atezolizumab adjuvant treatment. The patient is still under radiological follow-up to evaluate any disease relapse, and he is disease-free 5 months after diagnosis. Despite the inauspicious lymph involvement (pN3), surgery resulted in a valuable upfront treatment in this patient, and immunotherapy is going to consolidate as a valuable tool in a combined multidisciplinary approach to HAL, such as previously reported [11,31,36,45].

As of today, a double or triple regimen of platinum-based chemotherapy is the gold standard medical treatment in NSCLC despite cases of resistance to chemotherapy being registered [52]. This approach is also largely described in HAL probably because lack of diagnosis or unresectable masses and the scarcity of cases reported in the literature induce oncologists to treat this rare cancer as an NSCLC. This also happened to our two patients (Case 1 and Case 2).

At the basis of resistance to conventional therapies, in addition to acquired resistance mechanisms, a still uncertain role is played by cancer stem cells [53]. In this hierarchical model, the coexistence of different cell lineages should explain part of the therapy resistance. The neoplastic microenvironment is made by several mechanisms that induce the inefficiency of chemotherapies, causing tumor progression [54]. In addition to these tumor defense mechanisms, the change of histology after therapy is extremely rare but reported in the literature especially following the introduction of tyrosine kinase inhibitors. Adenocarcinomas (ADC) usually translate into small cell carcinomas (SCC) or in squamous cell carcinomas (SqCC). The main mechanism described is resistance to inhibition of the EGFR pathway. A recent case series described several histologic transitions from ADC to SCC [55]. Moreover, in EGFR mutated cohort resistant to chemotherapy, it was reported a transition to SCC in 14% of 37 NSCLC cases [56]. The microscopic features and the expression of typical markers such as HepPar1, aFP, and CK7 make sure the diagnosis of HAL. In this discussion, a possible histologic transition from adenocarcinoma to HepPar1+ primitive lung cancer must be further studied. As a matter of fact, Ishikura and colleagues [12] suggested that an aberrant histologic transition (from ADCs to HAL) could be possible, considering that the respiratory branch is originated by a median anterior gut diverticulum (common embryological origin of the respiratory and digestive tract), but further investigation on this point need to be addressed.

Another crucial point is related to the rarity of HAL: it is possible that this rare histology induces pathologists first to consider the neoplasm as NOS, prolonging the time to diagnosis and treatment decision-making (Case 2). Otherwise, it is possible to suspect a diagnosis (as the carcinoid in Case 3), finding HAL in the operative specimens. On this view, when feasible, surgery probably remains an interesting upfront choice in the early stages, and N0 patients will probably benefit from surgical treatment, despite the extremely malignant histology.

An emerging and interesting point is reserved for new therapies administrated in HAL. Anlotinib and Osimertinib, the latest novelties in TKI therapy, are ameliorating outcomes in advanced lung tumors [57]. As is well known, a particular TKI called Sorafenib is largely adopted in advanced hepatocellular carcinoma [58]. In the literature, few TKI experiences in HAL are available, sometimes with comforting results. Chen et al. achieved a 29-month survival in an advanced ill patient with an EGFR T790M mutation treated with Osimertinib and Anlotinib [30]. In 2015, a single case of HAL survived 11 months after diagnosis with the treatment of Sorafenib, in addition to conventional chemotherapy (a standard NSCLC platinum-based doublet scheme) [33]. Once again, Xu et al. recently reported a notable result in an unresectable, aFP-producing, bulky HAL with no response to multilinear therapies that demonstrated an improved outcome with Sorafenib (400 mg twice daily) and more than 13 months of survival [51]. Considering that the median overall survival in HAL is 14 months, these results must be taken into consideration. To note, Sorafenib has not been employed directly in HALs. It has been previously used in NSCLC as a novelty solution in terms of disease-free survival, improving disease control rate in selected patients with advanced lung cancers (EGFR wild type and aberrant expression of KRAS). Despite this, Sorafenib failed to highlight a clear benefit in terms of survival in advanced NSCLC, even though increased disease control has been achieved [59]. In the last few years, a growing experience of Sorafenib and other TKIs has been notable, and as a result, the role of TKIs in HepPar1+ NSCLC should be further investigated. In this complicated picture, emerging cases in the literature are going to clarify and delineate new aspects and trends of this cancer, improving knowledge and spacing on innovative fields, such as new therapies (TKIs).

## 6. Conclusions

As we summarized, HAL remains an unclear malignancy with a poor prognosis, and a multidisciplinary approach is mandatory. Free nodes at the early stages (N0) have a greater chance of surviving in HAL compared to N+. Although surgery does not always guarantee a better outcome, it remains an interesting and competitive option when it is feasible. Few promising experiences of monoclonal antibodies and TKIs are going to delineate a wider range of therapies in HAL that must be considered in the future.

As a main conclusion, we must consider suspect of HAL any patient presenting with a right upper lobe mass (and left lobes involvement in females) with a diameter of around 5 cm having no response to routine chemotherapy and an unclear preoperative diagnosis (NOS), especially in men. These cases must be carefully investigated because the scarce prognosis and an unshared consensus treatment make the time to diagnosis extremely valuable.

## Figures and Tables

**Figure 1 jcm-12-01411-f001:**
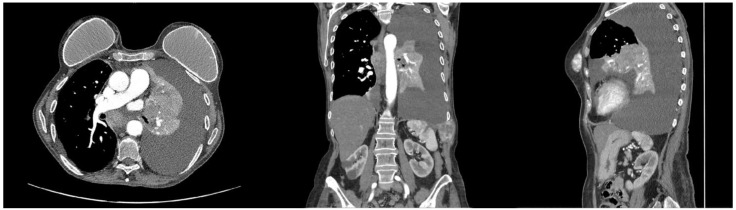
Representative CT-scan images of Case 1. The left apicobasal pleural effusion is seen obscuring the neoplasm. The lung parenchyma is collapsed. The subsequent palliative placement of a 20Ch drainage and talc pleurodesis guaranteed a sufficient lung expansion. No surgical options were available due to heavily compromised oncological status.

**Figure 2 jcm-12-01411-f002:**
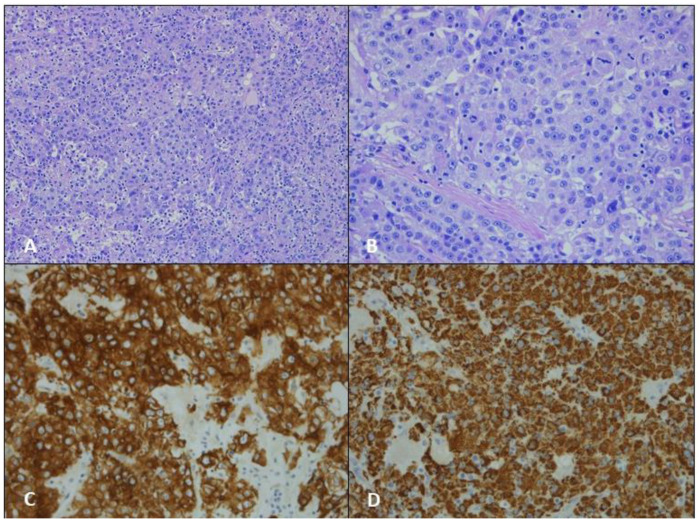
(**A**) Hematoxylin and eosin staining, original magnification ×50; (**B**) hematoxylin and eosin staining, original magnification ×100; (**C**) IHC: MNF116, original magnification ×100; (**D**) IHC: HepPar-1, original magnification ×100.

**Figure 3 jcm-12-01411-f003:**
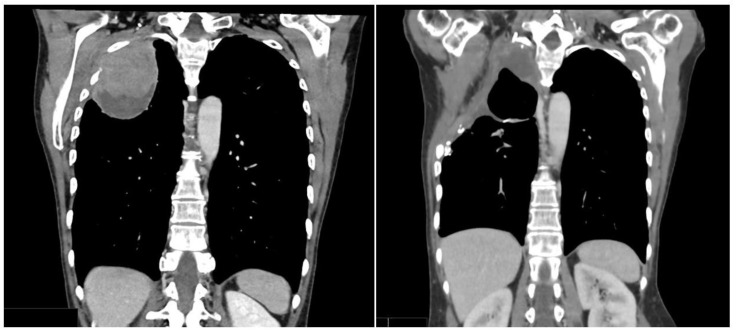
Representative CT-scan images of Case 2. A preoperative right upper lobe bulky mass is seen. Postoperative right upper lobectomy and second to fifth costectomy.

**Figure 4 jcm-12-01411-f004:**
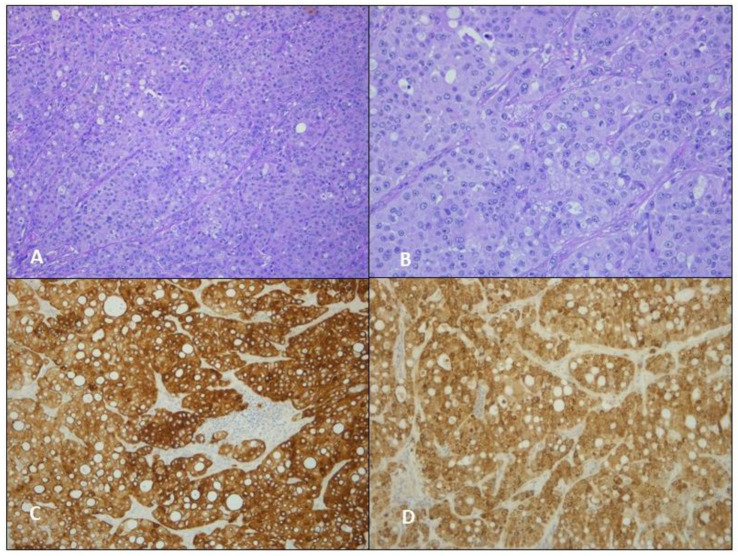
(**A**) Hematoxylin and eosin staining, original magnification ×50; (**B**) hematoxylin and eosin staining, original magnification ×100; (**C**) IHC: MNF116, original magnification ×100; (**D**) IHC: HepPar-1, original magnification ×100.

**Figure 5 jcm-12-01411-f005:**
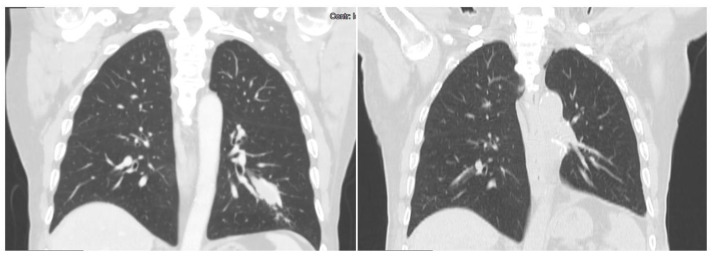
Preoperative and postoperative CT-Scan of Case 3. A left lower lobe mass is seen. Five months after lobectomy, no evidence of disease relapse is reported.

**Figure 6 jcm-12-01411-f006:**
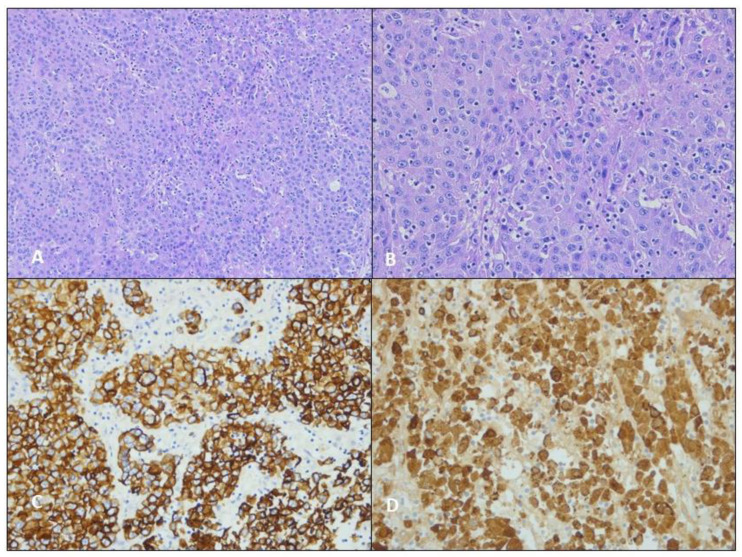
(**A**) Hematoxylin and eosin staining, original magnification ×50; (**B**) hematoxylin and eosin staining, original magnification ×100; (**C**) IHC: MNF116, original magnification ×100; (**D**) IHC: HepPar-1, original magnification ×100.

**Figure 7 jcm-12-01411-f007:**
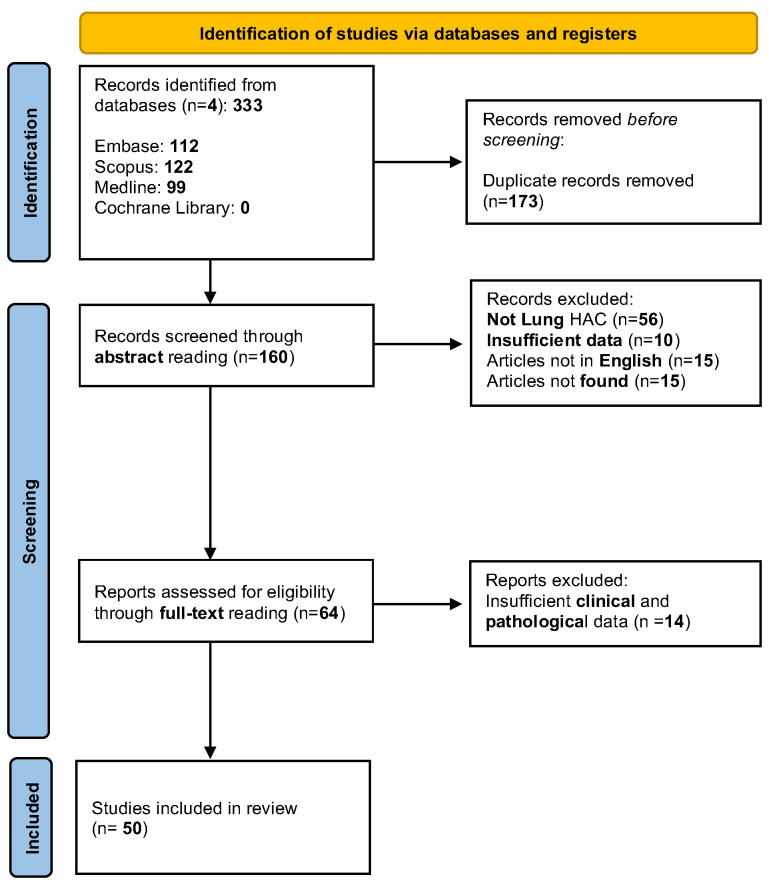
Included studies according to the PRISMA 2020 flow diagram [23].

**Figure 8 jcm-12-01411-f008:**
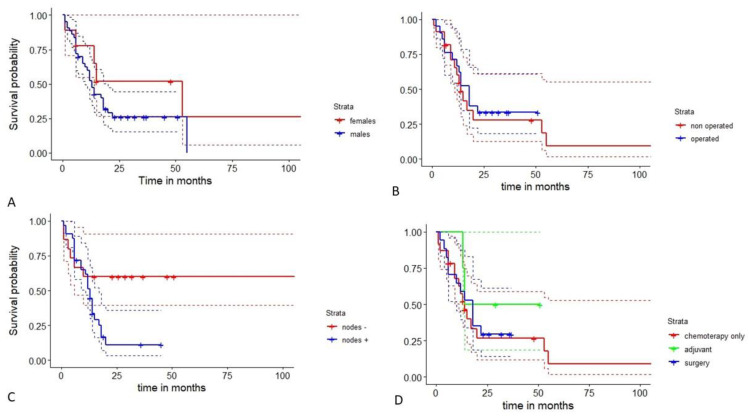
Survival probability in HAL patients considering gender: females appear to be protected (**A**). Surgery does not guarantee improved survival, but it seems to be quite longer than non-operated ones (**B**). Even though HAL has a well-known poor prognosis, early stages (N0) demonstrate a statistically improved survival, especially one year after surgery, according to the log-rank test (*p* = 0.04) (**C**). Survival appears to be similar in HALs treated with chemotherapy only and surgery, but adjuvant chemotherapy seemed to give an improved chance, despite little experience available (**D**).

**Table 1 jcm-12-01411-t001:** Immunohistochemical stains performed in the three cases of HAL.

ANALYSES	Case 1	Case 2	Case 3	ANALYSES	Case 1	Case 2	Case 3
MNF116	-/+	+	+	EGFR	wt	wt	wt
p40	-	-	-	ALK	wt	wt	wt
Syn	-	-	-	ROS	wt	wt	wt
CgA	-	-	-	RET	wt	wt	wt
Ki67	40%	80%	60%	MET	wt	wt	wt
TTF-1	-	-	-	K-RAS	wt	wt	wt
BRG1	+	+	+	BRAF	wt	wt	wt
CD34	-	-	-	Pan-TRK	NA	NA	wt
ER	-	-	NA				
PR	-	-	NA	PD-L1	3%	-	90%
HepPar-1	+	+	+				

NA = Not available, wt = wild type, “+”= positive, “-”= negative stain. Ki67 is a proliferation marker. BRG1 and HepPar1 were positive in all cases reported. At histological examination, tumor cells were large and polygonal, with abundant eosinophilic cytoplasma and central eosinophilic nucleoli. Immunohistochemical analysis showed a strong diffuse positivity for cytokeratin MNF116, HepPar-1 (Hepatocyte Paraffin 1). TTF-1, p40, napsin A, chromogranin (CgA), synaptophysin (Syn), CD56, CK20, CDX2, and CD34 were found to be negative. BRG1/SMARCA4 was retained.

**Table 2 jcm-12-01411-t002:** Demographics, pathological features, and treatments in HAL.

Variables	Results
Age (years) mean ± DS (range); median	59.7 ± 8.5 (33–79); 61
Sex (M/F) n(%)	57 (86.4)/9 (13.6)
Smokers n(%)	44 (66.7)
Alpha-fetoprotein (ng/mL) mean ± DS (range); median	13,543 ± 36,097 (<20–203,320); 902
Thoracic pain at diagnosis n(%)	25 (36.8)
Neoplasm surface (cm^2^)	38.1 ± 29.0 (4.4–132); 31.4
Side (right/left) n(%)	38 (64.4)/21 (35.6)
Lung field n(%)	
Upper lobe	41 (68.4)
Mid lobe	2 (3.3)
Lower lobe	12 (20.0)
Others (parahilar and paratracheal)	5 (8.3)
Stage n(%)	
I	5 (8.8)
II	5 (8.8)
III	23 (40.4)
IV	24 (42.1)
Lymph nodes involvement (yes/no) n(%)	40 (72.7)/15 (27.3)
Metastases involvement (yes/no) n(%)	23 (41.8)/32 (58.2)
Surgery (yes/no) n(%)	25 (50.0)/25 (50.0)
Lobectomy	21 (84.0)
Pneumonectomy	2 (8.0)
Wedge resection	2 (8.0)
Chemotherapy n(%)	
Chemotherapy only	26 (52.0)
Adjuvant	6 (12.0)
Chemotherapy protocol n(%)	
Double drug scheme	16 (50.0)
Triple drug scheme	6 (18.8)
Platinum n(%)	
Cisplatinum	10 (47.6)
Carboplatinum	9 (42.9)
Oxaliplatinum	2 (9.5)
Second drug n(%)	
Docetaxel	3 (13.6)
Etoposide	1 (4.5)
Gemcitabine	1 (4.5)
Paclitaxel	10 (45.5)
Pemetrexed	5 (22.7)
Uracil	1 (4.5)
Vinorelbine	1 (4.5)
Other first-line drugs n(%)	
Anlotinib	1 (9.1)
Bevacizumab	2 (18.2)
Crizotinib	1 (9.1)
Durvalumab	1 (9.1)
Erlotinib	1 (9.1)
Pembrolizumab	4 (36.4)
Sorafenib	1 (9.1)

**Table 3 jcm-12-01411-t003:** Surgical or medical proposed treatments in HAL.

Author	Year	Treatment	Surgery	Platinum-Based Scheme	Other Drugs	Savage Therapy or Other Regimens	Follow-Up (Time in Months)	Status
Al-Najjar et al. [13]	2015	CT		Cisplatinum, Docetaxel			12	Dead
Anusha G. et al. [26]	2021	CT		Carboplatinum, Pemetrexed			NA	NA
Ayub A. et al. [18]	2019	Surg	Lobectomy				6	Dead
Basse V. et al. [27]	2018	CT			Durvalumab		NA	NA
Chandan VS et al. [28]	2016	Surg	Lobectomy				NA	NA
		Surg	Lobectomy				NA	NA
Che YQ. et al. [29]	2014	CT		Cisplatinum, Paclitaxel		Nedplatin	20	Dead
Chen HF. et al. [30]	2019	Surg	Lobectomy	Cisplatinum, Pemetrexed		Osimertinib	29	Alive
Chen L. et al. [31]	2020	CT		Oxaliplatinum, Docetaxel		Pemetrexed, Oxaliplatinum, Bevacizumab	53	Dead
El Khoury A. et al. [32]	2019	CT		Cisplatinum, Etoposide	Pembrolizumab	Docetaxel, Nedaplatin	14	Alive
Gavrancic T et al. [33]	2015	CT		Carboplatinum, Paclitaxel	Sorafenib	Vinorelbine, Sorafenib	11	Dead
Haninger DM et al. [19]	2014	Surg	Wedge				37	Alive
		Surg	Wedge				10	Dead
Hayashi Y. et al. [34]	2002	Surg	Lobectomy				32	Alive
Hiroshima K. et al. [35]	2002	Surg	Lobectomy				12	Dead
		Surg	Sleeve lobectomy			Pneumonectomy	5	Dead
Hou Z et al. [15]	2021	CT		Cisplatinum, Pemetrexed		Anlotinib, Sorafenib	13	Dead
Khozin S. et al. [21]	2012	CT			Crizotinib		6	Alive
Kuan K. et al. [20]	2019	Surg	Lobectomy				4	Dead
Lagos G.G. et al. [36]	2021	CT		Carboplatinum, Paclitaxel	Pembrolizumab		7	Alive
Li J. et al. [7]	2019	CT			Anlotinib		6	Dead
Mokrim M. et al. [37]	2012	CT		Cisplatinum, Vinorelbine			7	Alive
Motooka Y. et al. [38]	2016	Surg	Lobectomy				51	Alive
Papatsimpas et al. [16]	2012	CT		Carboplatinum, Paclitaxel	Bevacizumab	Erlotinib	6	Dead
Qian GQ. et al. [39]	2016	CT			Erlotinib		1	Dead
Seddon J.E. et al. [40]	2021	CT		Carboplatinum, Pemetrexed	Pembrolizumab	Carboplatinum, Paclitaxel, RT	17	Dead
Shaib W. et al. [41]	2014	CT		Cisplatinum, Docetaxel			48	Alive
Shi YF. et al. [42]	2019	Surg	Lobectomy	Carboplatinum, Paclitaxel			14	Dead
Sun H. et al. [43]	2022	Surg	Lobectomy				36	Alive
		Surg	Lobectomy				26	Alive
		Surg	Lobectomy				18	Dead
		Surg	Lobectomy	Oxaliplatinum, Uracil			NA	NA
Sun J.N. et al. [44]	2016	Surg	Pneumonectomy				23	Alive
Terracciano et al. [45]	2003	Surg	Lobectomy				2	Dead
Tonyali O. et al. [5]	2020	Surg	Pneumonectomy	Carboplatinum, Paclitaxel		Nivolumab	14	Dead
Valle L. et al. [46]	2017	CT		Cisplatinum, Pemetrexed			55	Dead
Wang C. et al. [47]	2019	CT				Bevacizumab	9	Dead
Yang K. et al. [48]	2019	Surg	Lobectomy				18	Dead
Zhuansun Y. et al. [49]	2021	CT		Non platinum, Paclitaxel	Bevacizumab		9	Dead
		Surg	Lobectomy			Immunotherapy (Notspecified)	22	Dead
Chen, Ding et al. [10]	2022	Surg	Lobectomy	Cisplatinum, Paclitaxel			13	Dead
Zhen, Xiao-Chen et al. [50]	2022	CT		Carboplatinum, Paclitaxel	Pembrolizumab	Oxaliplatinum, Gemcitabine	13	Alive
Yao, Guan. et al. [51]	2022	Surg	Lobectomy				6	Dead
OUR CASES	2022	Surg	Lobectomy	Carboplatinum, Paclitaxel			5	Alive
		CT		Cisplatinum, Gemcitabine		Atezolizumab	15	Dead
		Surg	Lobectomy			Atezolizumab	5	Alive

## Data Availability

Not applicable.

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
