# Peer review of "Hepatoid Adenocarcinoma of the Lung: A Review of the Most Updated Literature and a Presentation of Three Cases"

_jcm, 2023, doi:10.3390/jcm12041411_

Round 1

Reviewer 1 Report (Previous Reviewer 1)

As the authors answers my first questions, i think that its still not attractive for readers and still no novelty here. especially if it was previously published since 1975 and the last one was between 2021-2022.

 im sorry but i still dont see anything interesting here

Author Response

Response to REVIEWER 1:

Point 1: As the authors answers my first questions, I think that its still not attractive for readers and still no novelty here. Especially if it was previously published since 1975 and the last one was between 2021-2022. im sorry but I still don’t see anything interesting here.

Response 1: Thanks to the Reviewer for the valuable comment. We are sorry once again because we did not reach the Reviewer’s interest, hoping to resubmit further studies able to involve the Reviewer’s curiosity. As he correctly discussed, Hepatoid Adenocarcinoma is a rare but well described disease in literature. Anyway, in our opinion this is not always a limit in reviews or case reports: in fact, we believe that different Authors could focus their research watching the same topic by several different points of view, increasing the scientific debate. This could be very useful especially in rare cancers, where evidence-based medicine is lacking, and new cases explain a larger and a stronger comprehension of the disease. Moreover, in our research we aimed to describe the state of art on treatments in HALs, that is still a not well-described aspect in literature.

Reviewer 2 Report (New Reviewer)

Review: Hepatoid adenocarcinoma of the lung: a systematic review of most updated literature and a single center experience.

Brief Summary:

The authors conduced a systematic review on HAL aiming to highlight proposed treatments available, comparing them in terms of survival. They also report three new female cases of HAL with clinical and pathological features.

General concept Comments:

This systematic review focused on the HAL and especially on the treatments opportunities is really interesting and brings compiled data that would be hard to find in the literature. It could be a precious help to clinicians for the rare subtype of cancer.

However, the main concern is that the cases brought on top of the literature by the authors are not sufficiently characterized to confirm the diagnosis of HAL. Especially because some have only been diagnosed after adjuvant treatment. Therefore, the authors should provides more evidences and results including a specific figure concerning the pathological diagnosis as all the classification of this rare cancer is based on this analysis.

It is necessary to improve English writing, but even more so typing.

Specific comments:

Line 12-13: “HAL is a rare lung cancer with immunohistochemical and microscopical features of hepatocellular carcinoma with no evidence of liver tumor.” The authors should also state that not only there shouldn’t be any tumor in the liver but also no lesion in any of the organs where hepatoid carcinoma can be found as stomach and ovaries are more frequent as primitive origin and can give rise to pulmonary metastases.

Line 35: “morphological and histopathological” What is the difference between those two terms for the authors? They should try to better determine what they think of when using the term “histopathological”. It usually refers to all the data generated through the pathological analysis including morphological, immunohistochemical and molecular data. In terms of HAL versus HCC only morphological and immunohistochemical data are shared but not the molecular characteristics.

Case 1 to 3 : As the major origin of hepatoid carcinoma arise from stomach and ovaries, the authors should provide all the data supporting the absence of lesion in any other organ than lung known to be a frequent origin for hepatoid carcinoma.

Case 1 and 2: As both patients received adjuvant treatment before the precise diagnostic of HAL has been made, the authors should confirm that the first samples made before the treatment are also characteristic of HAL (compatible morphology and immunochemical analysis typical of HAL with at least one or two hepatocellular markers expressed). Indeed, change of histology class is reported after treatment. That specific point should also be discussed in the appropriate section.

M&M: The authors should provide more information about how the pathological analysis was done as the diagnosis of HAL is only based on that, including how the immunohistochemical analysis was performed as well as the antibody references for all the ones discussed.

At least one figure should present the morphological and immunohistochemical characteristics of the patients added in this article.

Line 210: “Survival” The authors should precisely describe what type of survival they are talking about. If different survival types are being considered depending on the publications (OS, PFS,…) the authors should specify it and discuss this limit in the appropriate section.

Line 214: “an overall improved benefit is reported” The authors should clarify that sentence and better explain what kind of benefit is reported, how and by who.

Line 253: “HAL behave better than in males do”. The authors should adapt and temper this sentence based on the results as there is no statistical difference (stated in the sentence after this one).

Line 261: “dangerous”. Please change to aggressive

Author Response

Response to REVIEWER 2:

Point 1:  This systematic review focused on the HAL and especially on the treatments opportunities is interesting and brings compiled data that would be hard to find in the literature. It could be a precious help to clinicians for the rare subtype of cancer. However, the main concern is that the cases brought on top of the literature by the authors are not sufficiently characterized to confirm the diagnosis of HAL. Especially because some have only been diagnosed after adjuvant treatment. Therefore, the authors should provides more evidences and results including a specific figure concerning the pathological diagnosis as all the classification of this rare cancer is based on this analysis. It is necessary to improve English writing, but even more so typing.

Response 1: Thanks to the Reviewer for the valuable comment. As the reviewer suggested, we implemented parts of the cases 1, 2 and 3 reporting the anatomopathological diagnosis and adequate images of the cases. Anatomopathological diagnosis has been reported and specified in all its part.

We apologize for typing errors that we carefully corrected: we cannot understand why the words in the manuscript appeared stuck together.

Point 2:  Line 12-13: “HAL is a rare lung cancer with immunohistochemical and microscopical features of hepatocellular carcinoma with no evidence of liver tumor.” The authors should also state that not only there shouldn’t be any tumor in the liver but also no lesion in any of the organs where hepatoid carcinoma can be found as stomach and ovaries are more frequent as primitive origin and can give rise to pulmonary metastases.

Response 2: Thanks to the Reviewer for the valuable comment. We introduced in the text the valuable reviewer observation. Of course, during the preoperative examination, we made sure that non-liver neither other site of primitive or metastatic neoplasia were detectable.

Point 3: Line 35: “morphological and histopathological” What is the difference between those two terms for the authors? They should try to better determine what they think of when using the term “histopathological”. It usually refers to all the data generated through the pathological analysis including morphological, immunohistochemical and molecular data. In terms of HAL versus HCC only morphological and immunohistochemical data are shared but not the molecular characteristics.

Response 2: Thanks to the Reviewer for the valuable comment. According to the Italian pathological report, we refer as “morphological” the macroscopical description of the tumor (what the pathologist discover with no microscope need). Histopathological aspects refer to the specimens prepared to the microscopical analysis. This part is made by two sections: a first base-colored examination (for example hematoxylin and eosin) and then an immunohistochemically analysis (in this case HepPar1, TTF1, …). 

Point 4: Case 1 to 3: As the major origin of hepatoid carcinoma arise from stomach and ovaries, the authors should provide all the data supporting the absence of lesion in any other organ than lung known to be a frequent origin for hepatoid carcinoma.

Response 4: Thanks to the Reviewer for the valuable comment. As previously discussed, we provided all necessary information to disclaim this point in the manuscript.

Point 5: Case 1 and 2: As both patients received adjuvant treatment before the precise diagnostic of HAL has been made, the authors should confirm that the first samples made before the treatment are also characteristic of HAL (compatible morphology and immunochemical analysis typical of HAL with at least one or two hepatocellular markers expressed). Indeed, change of histology class is reported after treatment. That specific point should also be discussed in the appropriate section.

Response 5: Thanks to the Reviewer for the interesting and valuable comment. Change of histology after treatment is effectively reported in literature (especially from adenocarcinomas to squamous cell carcinomas or microcitomas) and we introduced a short paragraph in the discussion. Our cases were diagnosed as HAL after chemotherapy in two cases (Case 1 and 2). Because of the aggressive diagnosis, we processed the NOS specimens finding a positive HepPar1 stain in both cases. For this reason, we can confirm that HAL was the correct diagnosis. The rarity of HAL does not allow a first line routinary HepPar1+ research in all biopsies. The third case did not received adjuvant treatment. 

Point 6: M&M: The authors should provide more information about how the pathological analysis was done as the diagnosis of HAL is only based on that, including how the immunohistochemical analysis was performed as well as the antibody references for all the ones discussed.

Response 6: Thanks to the Reviewer for the valuable comment. We discussed with our pathologist this point providing an appropriate paragraph in the material section and reporting all the immunohistochemically processed stains in a table.

Point 7: At least one figure should present the morphological and immunohistochemical characteristics of the patients added in this article.

Response 7: Thanks to the Reviewer for the valuable comment. We discussed this point with our pathologists that provided appropriated pictures as attached in the manuscript.

Point 8: Line 210: “Survival” The authors should precisely describe what type of survival they are talking about. If different survival types are being considered depending on the publications (OS, PFS,…) the authors should specify it and discuss this limit in the appropriate section.

Response 8: Thanks to the Reviewer for the valuable comment. We must specify that in line 210 we considered the Overall Survival of the entire cohort (14 months). Considering gender, HAL returned a survival of respectively 13 months in males and 53 months in females.

Point 9: Line 214: “an overall improved benefit is reported” The authors should clarify that sentence and better explain what kind of benefit is reported, how and by who.

Response 9: Thanks to the Reviewer for the valuable comment. According to the K-M survival curve (Figure 4B) operated patients tend to live longer than non-operated ones, but significance is not reached. As the Reviewer highlighted, the sentence is not clear. We changed it as follows: “Surgery did not guarantee a significance in terms of survival, but operated patients demonstrated a 1-year and 3-years survival of 66.7% and 33.3 versus a 60.0% and 27.7% in non-operated ones respectively) (Figure 4B)”.

Point 10: Line 253: “HAL behave better than in males do”. The authors should adapt and temper this sentence based on the results as there is no statistical difference (stated in the sentence after this one).

Response 10: Thanks to the Reviewer for the valuable comment. We adapted the sentence as follows: “As we confirmed, female survival remains scarce, but survival in HAL tends to be longer in females compared to males”.

Point 11: Line 261: “dangerous”. Please change to aggressive

Response 11: Thanks to the Reviewer for the valuable comment. We corrected that non-academic sentence.

Reviewer 3 Report (New Reviewer)

Dear authors, 

I carefully reviewed your manuscript which combines a case presentation with a review of the literature regarding the HLA. Overall the manuscript is scientific sound and a review of the literature regarding this pathology can be of interest for some readers, although multiple reports are widely available, as you also found, therefore originality is limited. 

I have the following suggestions to make that I belive can improve the quality and attractivity of the manuscript: 

1. The title is overstating the manuscript content and purpose. It is not a single center experience, you are presenting only three cases from the last year and it is not a systematic review as you are not drawing any conclusions and you are not making any innovative analysis of the data. I suggest to name it simply as a review of the literature and a presentation of three cases. 

2. Abstract needs to be rephrased. I belive that you want to catch readers attention with the first phrase, but a more appropriate academic phrase should be used. 

3. Add a paragraph in the introduction where you detail the molecular landscape of these tumors, major genetic alterations and how they differ, or not, from classic LUADs.

4. Recheck manuscript for grammatical errors, example - ligne 62. "discussedtreatments

5. Please add pathology images for all the three cases including supporting immunohistochemistry, the diagnosis of HLA is made based on morphological and immunohistochemical profile.

6. Also, please add a paragraph in the discussion regarding the differential diagnosis of these tumors. 

7. Results section needs to be reorganised. It is not clear how you generated the data from Table 1, also is strange that you selected 50 articles from the literature but your cohort is around 60 cases in total. Please include an annex where we can check all the 50 articles included in the review. 

8. For Table 2 - also add the year when the study was published. This way is easier to see differences in the treatment regimens temporarily.

9. Please check all the refs cited in the manuscript for accuracy. A red sign is regarding ref 4 which refers to hepatoid adenocarcinoma of the stomach! 

Kind regards,

Author Response

Response to REVIEWER 3:

Point 1: The title is overstating the manuscript content and purpose. It is not a single center experience, you are presenting only three cases from the last year and it is not a systematic review as you are not drawing any conclusions and you are not making any innovative analysis of the data. I suggest to name it simply as a review of the literature and a presentation of three cases.

Response 1: Thanks to the Reviewer for the valuable comment. We adjusted the title as you suggested.

Point 2: Abstract needs to be rephrased. I belive that you want to catch readers attention with the first phrase, but a more appropriate academic phrase should be used.

Response 2: Thanks to the Reviewer for the valuable comment. We rephrased abstract and we ameliorated the first sentence as academic.

Point 3: Add a paragraph in the introduction where you detail the molecular landscape of these tumors, major genetic alterations and how they differ, or not, from classic LUADs.

Response 3: Thanks to the Reviewer for the valuable comment. As you suggested, we introduced a brief paragraph on molecular landscape and genetic hallmarks in the introduction.

Point 4:  Recheck manuscript for grammatical errors, example - ligne 62. "discussedtreatments "

Response 4: Thanks to the Reviewer for the valuable comment. As discussed with Reviewer 2, we seriously apologize for typing errors. We do not understand why the words in the manuscript appeared stuck together in the paper you reviewed.

Point 5:  Please add pathology images for all the three cases including supporting immunohistochemistry, the diagnosis of HLA is made based on morphological and immunohistochemical profile.

Response 5: Thanks to the Reviewer for the valuable comment. We discussed with our pathologists this point and we introduced the requested images.

Point 6: Also, please add a paragraph in the discussion regarding the differential diagnosis of these tumors.

Response 6: Thanks to the Reviewer for the valuable comment. We introduced a short paragraph with the differential diagnosis that clinicians must consider in the diagnostic process.

Point 7: Results section needs to be reorganised. It is not clear how you generated the data from Table 1, also is strange that you selected 50 articles from the literature but your cohort is around 60 cases in total. Please include an annex where we can check all the 50 articles included in the review.

Response 7: Thanks to the Reviewer for the valuable comment. We introduced an Appendix in which you can check all articles included. The cohort is larger because several case reports reported more than a single case. In Table 1 descriptive include all data available. As you know, collecting different case report, reported information could be different and not always clear or complete. We tried to complete information as much as possible.

Point 8: For Table 2 - also add the year when the study was published. This way is easier to see differences in the treatment regimens temporarily.

Response 8: Thanks to the Reviewer for the valuable comment. We introduced the publication data in the first column.

Point 9: Please check all the refs cited in the manuscript for accuracy. A red sign is regarding ref 4 which refers to hepatoid adenocarcinoma of the stomach!

Response 9: Thanks to the Reviewer for the valuable comment. We re-checked all the listed references. As you correctly underlined, ref.4 concerns a HAC of the stomach, but we used it to the pathological definition and to report the prevalence of this malignancy in the most common involved site of HAL. We corrected the position of the citation in the draft.

Reviewer 4 Report (New Reviewer)

Dear authors,

very interesting subject and well written paper!

only some minor english mistakes

Author Response

Response to REVIEWER 4:

 Point 1: Dear authors, very interesting subject and well written paper! only some minor english mistakes

Response 1: Thanks to the Reviewer for the valuable comment. We are very satisfied for meeting the interesting of the reviewer. We reviewed the suggested English mistakes.

Reviewer 5 Report (New Reviewer)

Thank you for the opportunity to review a manuscript entitled “Hepatoid adenocarcinoma of the lung: a systematic review of 2 most updated literature and a single center experience”. The manuscript presents three cases of hepatoid adenocarcinoma and provides the results of a systematic review focused on the characteristics and treatment options of this diagnosis.

Overall, the manuscript is well written. The work is well described in all details. The topic is correctly introduced, methods are correctly chosen, and the findings are clearly presented and sufficiently discussed.

I have found only few details:

Line 62 – “we reported three more cases of HAL in females” But only two cases were female, and one was male.

I would also recommend adding a few more findings from the review into the Abstract, e.g., the predominance in the right upper lobe.

Finally, a language correction along with minor formatting mistakes need to be addressed.

Author Response

Response to REVIEWER 5:

Point 1: Thank you for the opportunity to review a manuscript entitled “Hepatoid adenocarcinoma of the lung: a systematic review of 2 most updated literature and a single center experience”. The manuscript presents three cases of hepatoid adenocarcinoma and provides the results of a systematic review focused on the characteristics and treatment options of this diagnosis. Overall, the manuscript is well written. The work is well described in all details. The topic is correctly introduced, methods are correctly chosen, and the findings are clearly presented and sufficiently discussed.

Response 1: Thanks to the Reviewer for the valuable comment. We are very pleased to have met the Reviewer’ interest.

Point 2: Line 62 – “we reported three more cases of HAL in females” But only two cases were female, and one was male.

Response 2: Thanks to the Reviewer for the valuable comment. We corrected the sentence as you suggested.

Point 3: I would also recommend adding a few more findings from the review into the Abstract, e.g., the predominance in the right upper lobe.

Response 3: Thanks to the Reviewer for the valuable comment. We rephrased the abstract, better focusing on the main HAL aspects we described and discussed in the manuscript.

Point 4: Finally, a language correction along with minor formatting mistakes need to be addressed.

Response 4: we reviewed English and all the sentences that sound difficult to understand or which were far from the current academic average speaking.

Round 2

Reviewer 1 Report (Previous Reviewer 1)

looking better than it was before with more additional informations

Reviewer 3 Report (New Reviewer)

Good work on improving the previous review round comments. I think that the manuscript is drastically improved. 

This manuscript is a resubmission of an earlier submission. The following is a list of the peer review reports and author responses from that submission.

Round 1

Reviewer 1 Report

Hepatoid adenocarcinoma is known to be a rare disease, but it was reported several times in the literature, i searched and found more than 9 as written in the article, A total of 65 Hepatoid adenocarcinoma were identified from the database, including 31 male and 34 female patients. that what was found in article that was published a year a go.

no novility here.

Reviewer 2 Report

This manuscript reports two cases of hepatoid adenocarcinoma of the lung (HAL) and reviews the different chemotherapy protocols, female cases of HAL, as well as the survival probability in HAL of male and female. As HAL is an extremely rare malignant tumor, cases reports are valuable for the treatment and research. However, if as case report, this manuscript is so primary and simple, lack of diagnosis evidence such as pathological confirmation, immunohistochemical staining results, genetic test results, and treatment details. As a review, it lists limited literatures for support the author’ opinion. Even I could not draw what is the main idea the authors wanted to describe from their review part if the discussion part could be regarded as review.